# A Pilot Serosurvey for Selected Pathogens in Feral Donkeys (*Equus asinus*)

**DOI:** 10.3390/ani10101796

**Published:** 2020-10-02

**Authors:** Erin L. Goodrich, Amy McLean, Cassandra Guarino

**Affiliations:** 1Cornell University College of Veterinary Medicine, Animal Health Diagnostic Center, Ithaca, NY 14853, USA; cg82@cornell.edu; 2Department of Animal Science, University of California, Davis, CA 95616, USA; acmclean@ucdavis.edu

**Keywords:** *Borrelia burgdorferi*, burro, donkey, equid, equine herpesvirus, equine influenza, West Nile virus

## Abstract

**Simple Summary:**

This study aimed to assess the pathogen exposure status of recently captured feral donkeys from Death Valley National Park, California. Assays to detect the presence of antibodies to equine herpesvirus 1, equine influenza virus, West Nile virus, and *Borrelia burgdorferi* (the causative agent of Lyme disease) were performed on serum samples from these feral donkeys. The results indicate that this population is mostly naïve and likely susceptible to these common equid pathogens upon removal from the wild.

**Abstract:**

Recent removal and relocation of feral donkeys from vast public lands to more concentrated holding pens, training facilities, and offsite adoption locations raises several health and welfare concerns. Very little is known regarding the common equid pathogens that are circulating within the feral donkey population in and around Death Valley National Park, California, USA. The aim of this study was to utilize serologic assays to assess previous exposure of these donkeys to equine herpesvirus 1 (EHV-1), equine influenza (EIV), West Nile virus (WNV), and *Borrelia burgdorferi* (the causative agent of Lyme disease). The results of this study indicate that this feral equid population is mostly naïve and likely susceptible to these common equid pathogens upon removal from the wild.

## 1. Introduction

The feral donkey (*Equus asinus*), or burro, was introduced to North America as a result of Spanish colonization in the sixteenth century. Being well adapted to thriving in desert environments, and useful as a mode of transportation for people and equipment, these donkeys later became utilized by miners in geographical regions that would later become United States of America’s (US) National Parks, such as Death Valley National Park [1]. As is the case with many introduced species, feral donkeys are the subject of much debate regarding the perceived associated positive and negative ecological impacts. Some studies allege these feral burros are problematic due to competition with native animals for limited resources, changes to vegetation, damage to soils, and negative impacts on springs [2,3]. Other researchers have studied the feral burros of the Sonoran Desert and found beneficial ecological impacts where they are noted to dig deep groundwater wells that are subsequently also utilized by several other species of mammals and birds [3,4]. One strategy employed to maintain appropriate management levels by both the US Department of the Interior’s Bureau of Land Management (BLM) and the United States Department of Agriculture’s Forest Service is removal of excess donkeys from public rangelands. Many of these donkeys subsequently become available for adoption into private care, while others go on to live in BLM-managed off-range facilities. Free-roaming donkeys in resource-limited environments, such as Death Valley, tend to spend their lives in small groups or pairs, or sometimes even in solitude, unlike the large herd formations that are typical of other equines in the wild [5]. Capturing these donkeys from the wild and placing them in groups with other unfamiliar animals likely produces some stress. This combination of stress and co-mingling creates an increased potential for viral reactivation (as in the case of equine herpesvirus 1 (EHV-1)) and pathogen transmission for communicable diseases such as EHV-1 and equine influenza (EIV) [6]. Moreover, there is a scarcity of scientific literature available regarding the common pathogen exposures and subsequent humoral immune responses of these feral donkeys in the wild, which has a direct impact on their risk of developing clinical disease upon capture, co-mingling, and relocation. Nasal samples were taken from this same population of donkeys to detect viral and bacterial pathogen levels using quantitative real-time PCR (qPCR). The study found little evidence of EHV1, 4, EIV, or *Streptococcus equi* subspecies *equi*, but the samples did contain DNA from asinine herpes virus 2, 3, and 5 and *Streptococcus equi* subspecies *zooepidemicus* [7].

Due to the vast land area, the desert climate, and the tendency for donkeys to roam in small groups or even alone, we hypothesized that the feral donkeys recently captured from Death Valley National Park would be naïve to many common pathogens that circulate in equid populations throughout the United States, including equine herpesvirus 1 (EHV-1), equine influenza (EIV), West Nile virus (WNV), and *Borrelia burgdorferi* (the causative agent of Lyme disease). The viral pathogens were selected as they are all commonly circulating among our domestic equids in the US and elsewhere, with the potential to cause significant clinical disease, and all have readily available serological assays to indicate previous exposure [8,9,10,11]. *B. burgdorferi* was selected, as it is a common infection in horses living in endemic areas, and the geographic range of the vector continues to expand across the United States [12].

EHV-1 and EIV are both contagious pathogens capable of being transmitted among equids, especially those in close proximity. Likewise, equid populations commonly demonstrate exposure to WNV and *B. burgdorferi*; however, these two pathogens require competent vectors for transmission. Commercial equine vaccines are available for EHV-1, EIV, and WNV, although they have not been evaluated for use in donkeys. Likewise, commercial *B. burgdorferi* vaccines exist for use in dogs; however, some research suggests they may offer protection against infection in horses as well, although antibody response may be short-lived, and more frequent boosters may be required [13,14]. Our objective was to assess antibody levels to these common equid pathogens in recently captured feral burros from Death Valley as a means of understanding their risk for disease development and transmission. The results from this study can be used to inform protocol design surrounding the handling of feral donkeys upon removal from the wild, especially with regard to preventive medicine, vaccination practices, and movement to other geographical regions. 

## 2. Materials and Methods

Blood was collected from 98 feral donkeys removed from the Death Valley National Park range (Shoshone, CA, USA) on three separate occasions. The first set of samples was collected in early November 2018 from 51 donkeys that were captured within 10 days. A second (*n* = 35) cohort was sampled in late November, and a third (*n* = 12) was sampled in December before leaving short-term holding. Each of the animals in these later two groups was sampled within 5 days of capture. Donkeys were co-mingled with adult females and foals held together and separated from the adult male group. A 10 mL blood sample was collected by venipuncture of the jugular vein. All animals were observed by veterinarians prior to sampling. Body condition score (BCS) on a 5-point scale [15], sex (*n* = 49 males, *n* = 49 females), and approximate age were recorded based on that veterinary evaluation. The protocol was approved by the University of California Davis Institutional Animal Care and Use Committee #20611.

The equine herpesvirus type 1 risk evaluation assay (Animal Health Diagnostic Center (AHDC), Cornell University, Ithaca, NY, USA) was performed as previously described [16]. Briefly, a fluorescent bead bound to an EHV-1 recombinant protein, glycoprotein C (gC), was incubated for 30 min with serum diluted 1:400 in phosphate buffered saline ((PBN) with 1% (w/v) bovine serum albumin (BSA) and 0.05% (w/v) sodium azide). Multiscreen®HTS plates (Millipore, Danvers, MA, USA) were used for incubations, wash steps were performed with phosphate buffered saline containing 0.05% (*v*/*v*) Tween 20 (PBS-T) using an ELx50 plate washer (Biotek Instruments Inc, Winooski, VT, USA), and all incubations were performed at room temperature. Serum antibodies bound to beads were detected with either biotinylated goat anti-horse IgG (H+L) antibody (Jackson Immunoresearch Laboratories, West Grove, PA, USA) or biotinylated anti-equine IgG4/7 (RRID: AB_ 2820277), incubated for 30 min. Streptavidin-PE (Invitrogen™, Carlsbad, CA, USA), incubated for 30 min, was used to label the detection antibody. Finally, beads were suspended in PBN and analyzed in a Luminex IS 100 instrument (Luminex Corp, Austin, TX, USA). The data were reported as median fluorescent intensities (MFI). The two values obtained from the assay were used to categorize animals as “high risk”, “moderate risk”, “low risk”, and “very low risk” of contracting EHV-1 if exposed to the virus [16].

The equine influenza virus (EIV) hemagglutination inhibition assay (HAI) assay (AHDC, Cornell University, Ithaca, NY, USA) was performed as previously described [17], using an equine influenza strain isolated in New York in 2018, A/Eq/NY/192142/18 (H3N8). Briefly, serum samples were heat-inactivated at 56 °C for 30 min and treated with 0.016M potassium periodate solution, then subsequently neutralized with glycerol to prevent nonspecific reactions. Nonspecific hemagglutinins were removed by pre-incubation with turkey red blood cells for 30 min. Two-fold serial dilutions of each prepared serum, beginning at 1:4, were incubated with constant concentration of virus for 30 min at room temperature, and turkey red blood cells were then added. Following 30–45 min incubation at room temperature, wells were examined for the presence or absence of viral induced hemagglutination. The antibody titer of the serum was reported as the reciprocal of the last dilution of serum, which completely inhibited hemagglutination. Samples were run in duplicate with half titers reported when the end point was between two dilutions. Positive, negative, and serum sample controls were run concurrently. Historic data from diagnostic submissions requesting EIV HAI serologic testing (AHDC, Cornell University), totaling 916 equine samples submitted from 2013 to 2017 and 18 Equidae samples submitted from 2012 to 2020 were used to compare to expected results in domestic populations. 

The West Nile virus (WNV) antigen capture IgG and IgM ELISAs (AHDC, Cornell University, Ithaca, NY, USA) were performed as previously described [18], with anti-equine IgM (RRID:AB_2737323) or anti-IgG1/3 (RRID:AB_2737325) as capture antibody, and biotinylated anti-WNV E-protein (RRID:AB_2744504) as the detection antibody. Briefly, Nunc MaxiSorp™ plates (Invitrogen™, Carlsbad, CA, USA) were coated overnight at 4 °C with 4 µg/mL of the capture antibody in Carbonate buffer, pH 9.65. Plates were washed between each incubation step with phosphate buffered saline containing 0.05% (*v*/*v*) Tween 20 (PBS-T). Each test and control sample was diluted 1:100 in tris-buffered saline containing 0.05% (*v*/*v*) Tween 20 (TBS-T) with 5% (w/v) nonfat dry milk and incubated for 1 h at 37 °C. Cell-culture-derived WNV antigen and control antigen were added to the positive and negative wells of each sample, respectively, and incubated overnight at 4 °C; inactivated WNV antigen was prepared from Verocells infected with a chimeric WNV [19], and control antigen was prepared from non-infected cells. Detection antibody was incubated for 1 h at 37 °C, followed by streptavidin-peroxidase (Jackson ImmunoResearch, West Grove, PA, USA) for 1 h at 37 °C. Plates were developed with TMB (KPL, Gaithersburg, MD, USA). Results were reported as a positive/negative (P/N) ratio—the optical density (OD) of the WNV antigen wells, divided by the OD of the negative antigen wells. 

The Lyme Multiplex assay (AHDC, Cornell University, Ithaca, NY, USA) was performed as previously described [20]. Briefly, three sets of fluorescent beads bound to recombinant OspA, OspC, and OspF proteins, respectively, were incubated for 30 min with serum diluted 1:400 in PBN. Multiscreen HTS plates (Millipore, Danvers, MA, USA) were used for incubations, wash steps were performed with PBS-T using an ELx50 plate washer (Biotek Instruments Inc, Winooski, VT, USA), and all incubations were performed at room temperature. Serum antibodies bound to beads were detected with biotinylated goat anti-horse IgG (H + L) antibody (Jackson Immunoresearch Laboratories, West Grove, PA, USA), incubated for 30 min. Streptavidin-PE (Invitrogen™, Carlsbad, CA, USA), incubated for 30 min, was used to label the detection antibody. Finally, beads were suspended in PBN and analyzed in a Luminex IS 100 instrument (Luminex Corp, Austin, TX, USA). The data were reported as median fluorescent intensities (MFI), and cut-off values for “negative”, “equivocal”, and “positive” were determined based on values from horse serum [20]. Historic data from diagnostic submissions requesting *B. burgdorferi* serologic testing on serum (AHDC, Cornell University) totaling 5468 equine samples submitted the summer of 2018 and 1271 Equidae samples submitted 2011 to 2020 were used to compare expected results in domestic populations.

## 3. Results

Approximate ages of the 98 donkeys removed from the Death Valley National Park ranged from 4 months to 20 years, with a median of 6 years (48 females and 48 males). Of the 70 animals for which BCS data were approximated, two were <2, and the remaining 68 had a mean BCS of 3.08, with a median of 3.0 on a scale of 1–5 [21]. Results of the serologic assays are summarized in Table 1. 

A total of 92 serum samples were tested in the EHV-1 risk evaluation assay. In total, 3% (3/92) of the donkeys that were tested had antibody values that would categorize them as “low risk” for infection with EHV-1, 32% (29/92) demonstrated evidence of exposure to an equine herpes virus and would be categorized as “moderate risk” for infection with EHV-1, and the remaining 65% (60/92) of the burros were categorized as “high risk” for EHV-1 infection at the time of sampling.

Sufficient serum was available for testing 70 of the donkeys for evidence of exposure to EIV. Equine influenza H3N8 antibody titers were not detected at a minimum dilution of 1:4 in 13% (9/70) of donkeys, the highest titer of 12 was found in 13% (9/70) of donkeys, and the remaining 74% (52/70) of donkeys had a titer of 6 (Figure 1). Of the 18 historic domestic Equidae submission, only two samples had a titer of <8. 

All of the donkeys tested negative for both IgM and IgG antibodies to West Nile virus. In total, 7% (6/92) of the donkeys tested positive for at least one of the three antibodies to *B. burgdorferi*. Five of those animals were positive for antibodies against the chronic infection marker, OspF, and one donkey was positive for only the early infection marker, OspC. One additional donkey had an “equivocal” value for OspF. In the domestic equidae submissions, 48% (616/1271) of animals tested positive for the infection markers OspC and/or OspF, compared to only 27% (1480/5468) of domestic equines.

## 4. Discussion

Our hypothesis was that equids removed from Death Valley National Park would have had limited contact with common infectious disease agents that are frequently found in equid populations in the USA and would be serologically naïve. The serologic survey conducted in this study supports this hypothesis.

A significant portion of this population of donkeys was found to be at high risk for infection with EHV-1, a common pathogen ubiquitous in most equine populations. EHV-1 can affect all members of the *Equidae* family, including donkeys. Infection with EHV-1 can manifest as a range of clinical presentations, from mild respiratory illness, to neurologic sequelae that can be fatal, and EHV-1 infection of pregnant mares during gestation can lead to late-term abortion [9,22,23,24]. EHV-1 typically results in a latent infection with subsequent re-activation and associated clinical disease and viral shedding during periods of stress, such as traveling and co-mingling [9,23]. A limitation of this testing is the specificity for EHV-1; while infection with asinine herpesvirus 3 (EHV-8) or EHV-9 might generate cross-reactive antibodies based on homology to the target antigen in the EHV-1 assay, evidence of infection with other herpesvirus strains, including EHV-4, might not be detected. However, asinine herpesvirus 2, 3, and 5 were detected in nasal swabs using qPCR in this population of donkeys [7]. Numerous vaccines are commercially available to protect horses and other equids from EHV-1 infection, including killed virus vaccines, a live virus vaccine, and formulations approved for use during pregnancy to prevent abortion, although there are no current vaccine claims to protect against the neurologic form of disease [25].

EIV infection in horses typically causes acute febrile respiratory disease, which can frequently be complicated by secondary bacterial infection, especially in unvaccinated animals [26]. EIV infection in donkeys can cause pyrexia, cough, nasal discharge, and lethargy, similar to the signs seen in infected horses [24]. However, donkeys appear to be more susceptible than horses to infection with EIV and to developing more severe clinical disease, including secondary bronchopneumonia [27,28]. In 1963, an H3N8 strain of influenza was identified as the cause of a major epidemic in equines in the state of Florida, and in the following years, it spread throughout the USA and Europe [29]. The virus used in the EIV HAI assay was a Florida clade 1 virus isolated in NY that matches isolates recently found in the USA [30]. EIV HAI result titers below 8 are typically considered negative, indicating high risk for infection if exposed to EIV, and titers of 12 are low positive. All donkeys in this study, including the few donkeys with low positive EIV HAI titers, would likely be completely susceptible to infection with equine influenza if exposed. One limitation of this assay is the specificity for the H3N8 strain—if a novel strain were circulating in these donkeys, this assay may not detect those antibodies; however, these animals would likely still be susceptible to the commonly circulating strains in the USA. Numerous EIV vaccines are commercially available in the USA, including a modified live vaccine, a canary pox vector vaccine, and inactivated virus vaccines [31]. 

WNV has been endemic in North America since 1999 and can be transmitted to susceptible hosts through the bite of an infected *Culex* mosquito [32]. Birds typically serve as the predominant reservoir for WNV, and humans and equids are considered dead-end hosts [33]. Although only about 10% of unvaccinated, exposed equids usually develop associated neurologic disease, the clinical manifestation can be quite severe, including fever, ataxia, weakness, recumbency, muscle fasciculations, and death, with a case fatality rate of 30–40% [34,35,36,37]. WNV infection is diagnosed by identifying WNV-specific IgM in horse serum [38]. Following infection or vaccination, horses will develop WNV-specific IgG [18]. None of the donkeys in this population had evidence of previous exposure to WNV and could be completely susceptible to infection if transported to a region where WNV infected mosquitos are present. Numerous vaccines are currently available to protect horses and other equids from WNV infection [39]. 

In North America, *B. burgdorferi* is the most common cause of Lyme disease, with Ixodid ticks serving as the vector for transmission [12]. Clinical signs of Lyme disease in equids are not well described owing to the fact that cause and effect are very difficult to document; however, stiffness, lameness, hyperesthesia, and behavior changes are often described [40]. The associated clinical syndromes that have been well described in equids, although rare, include neuroborreliosis, uveitis, and cutaneous pseudolymphoma [12]. Seroprevalence studies suggest that *B. burgdorferi* exposure among equids is common throughout many regions of the United States and that the range is likely increasing [12]. In addition, these studies indicate that it is common for clinically normal horses living in endemic regions to have antibodies against *B. burgdorferi* [12]. Given the sensitivity of *Ixodes* ticks to desiccation [41] and the rugged, arid environment of Death Valley, it was surprising to find that a small number of donkeys had evidence of infection with *B. burgdorferi*. However, *B. burgdorferi* has been identified in ticks in areas of the Mojave desert surrounding Death Valley [42]. There is no significant information available regarding the susceptibility of donkeys to Lyme disease; however, the high seroprevalence in the diagnostic submissions from domestic Equidae suggests that these wild donkeys will be at risk for infection with *B. burgdorferi* if they are moved to a Lyme endemic area. There is not currently an approved vaccine to protect against infection with *B. burgdorferi* in horses; however, in endemic areas, some clinicians will use a canine Lyme vaccine off-label [14].

Throughout this study, comparisons were made utilizing historic diagnostic data, which have limitations. These limitations in some instances include unknown clinical history provided to the AHDC, and testing for specific indications which may include clinical disease or to measure vaccine response. The comparison serologic data in this study were not sorted in any way to account for these limitations. In addition, this study is not a comprehensive serologic survey of all possible equid pathogens; however, the results of this study and one other study do indicate that these wild equid populations may be extremely naïve and susceptible to many common pathogens upon removal from the wild, and there appears to be an increase in pathogen detection upon introduction to long-term holding facilities [7].

## 5. Conclusions

Given the fact that donkeys in the wild tend to remain in pairs or very small groups and that they often live in vast arid landscapes, the findings in this study may be applicable to feral donkeys in similar locations. The apparently naïve nature of these donkeys to common pathogens supports the need for exceptional care and biosecurity measures upon removing wild equids from their native habitat. Transportation vehicles and holding pens should be appropriately cleaned prior to movement in order to minimize risk of exposure to common equine pathogens. Further work is needed to understand the incidence of clinical disease from these pathogens in these populations upon removal from the wild, and co-mingling, in order to determine the optimal vaccination recommendations. 

## Figures and Tables

**Figure 1 animals-10-01796-f001:**
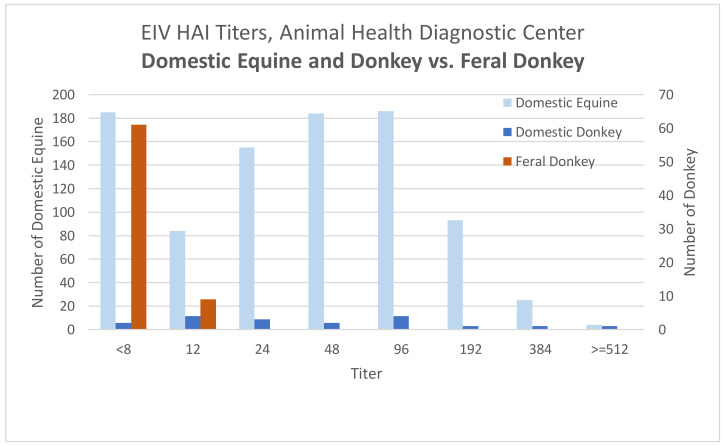
Equine Influenza Virus Hemagglutination Inhibition Titers (EIV HAI). Results for 70 feral donkeys from Death Valley in November and December, 2018, are compared to those obtained for over 900 domestic equines and Equidae (vaccinated and unvaccinated) tested at the AHDC, Cornell University, Ithaca, NY from 2013–2018.

**Table 1 animals-10-01796-t001:** **Summary of Serologic Assay Results**. The number of animals with serum antibodies that were reactive, moderately reactive, or non-reactive against each of the pathogens investigated.

Item	EHV-1	EIV	WNV	Lyme
**Reactive**	3	0	0	6
**Moderately Reactive**	29	9	0	1
**Non-Reactive**	60	61	92	85

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
