# Peer review of "A Pilot Serosurvey for Selected Pathogens in Feral Donkeys (Equus asinus)"

_animals, 2020, doi:10.3390/ani10101796_

Round 1
Reviewer 1 Report
The manuscript gives a valuable overview over the susceptibility of ferral donkeys to common equine pathogens. This concise article thus gives guidance on how to manage ferral donkeys caught for adoption or relocation.
The paper is well-written, clearly structured and provides sound data. The comparison of the results in ferral donkeys is compared to results in domestic horses. This could be improved by comparing the results to data from domestic donkeys.
Overall, the maniscript holds valuable new information relevant to the veterinary practitioners.
Author Response
Thank you for your comments.
To improve upon our comparison, we have included more historic data from the AHDC that incorporates results from domestic donkeys that we have tested in recent years.
Reviewer 2 Report
In their manuscript the authors conducted a pilot serological survey of the presence of specific antibodies to different pathogens in recently captured feral donkeys. The largely naïve status of these animals suggests their susceptibility to these pathogens. Although the manuscript is well-written, some important details are missing.
Major comments
- The authors should clearly explain why they picked these pathogens for the survey. Is there evidence of infection among domestic donkeys? For example, H3N8 equine influenza has been detected in donkey farms (i.e. PMID: 32341914, PMID: 32891950 etc). It would be beneficial if the authors did a deep literature search of the susceptibility of domestic donkeys to
- This study is about serologic assay, and this should be reflected in the title. Presumably the same donkeys were surveyed for bacterial and viral pathogens in another study, where PCR assay was used to detect viral RNA and bacterial genomic material (PMID: 32585994 – a study with the same IACUC approval number). Surprisingly, the authors didn’t refer to this recently published work in their manuscript.
- Figure 1 demonstrates immune status of domestic equines as positive controls. However, there is no mention of these animals in the Materials and Methods section. In addition, the figure is difficult to understand. What do the left and right Y axes mean?
- It would be beneficial if the authors added a summary table with the results of serosurvey for all tested pathogens.
Minor comments
- Lanes 34-36. The program names are not informative in this type of manuscripts. These programs can be cited in Materials and Methods, if relevant.
- Lanes 95-96. The HAI assay was performed on 1:4 diluted serum samples, with 2-fold further dilutions. It is not clear why Fig.1 shows titers 12, 24, 48 etc. If dilutions start from 1:4, it should be 4, 8, 16, 32, 64 etc.
- Lane 131: “serum samples”.
- Lane 137: not viral titers – should be antibody titers
- Lanes 174, 175, 177 – add spaces between EIV and HAI.
Author Response
Thank you for your comments.
Major revisions:
- Please see lines 50-54 for further clarification (and references) to explain our reasoning for focusing on these particular pathogens.
- We have called it a "Serosurvey" to reflect the fact that we used serologic assays in this study. Also please see lines 42-45 where we've included a statement regarding the PCR study. This study was also cited in lines 176-177.
- See lines 100-104 of the materials and methods section, which now describes the data in Figure 1. Also, please see revisions made to Figure 1.
- We have included a Table 1. that summarizes the serologic assay results.
Minor revisions:
All minor revisions have been addressed.
2. Please see lines 100-102 for an explanation of the method used when reporting out the EIV HAI titers.

Reviewer 3 Report
I think that this is a really interesting study, and a research area that has been neglected, so the authors are to be applauded for carrying out this work.
The manuscript is clear, concise and well written.
I only really have a couple of minor comments. I wonder if the conclusion is too strong and whether there is enough evidence to back this up? Although serological results are suggesting that donkeys are naive and therefore susceptible to these common pathogens - is there any evidence to suggest that infection with these pathogens is common in this population once removed from the wild? Given how many are routinely removed there must be some clinical data, even if only anecdotal, regarding incidence of clinical disease from these pathogens?
I think this would really be worth commenting on, even if it is just to acknowledge that this information is important but currently unknown. Perhaps that could be the next stage in this work?
I think without that, to say that 'equids should be vaccinated with all core vaccines as soon as possible following capture.' is probably a bit strong? Changing this even just to 'it is advised that equids that are removed and rehomed are vaccinated'? Or perhaps even better, something along the lines of 'further work is needed regarding the incidence of clinical disease from these pathogens in these populations, to determine optimal vaccination recommendations'.
This might seem pedantic, but the reason for raising this is that this recommendation could place alot of pressure on government agencies (or rehoming centres?) to be responsible to vaccinating these donkeys, which could add a significant financial burden to the management program, potentially to the detriment of more important aspects perhaps. Particuarly if a significant number remain in holding pens, are they genuinely at a high risk of infection with these pathogens? Surely there must be some clinical info about the donkeys in these facilities? Obviously if it is known that there is a high incidence of disease from these pathogens developing after removal from the wild, then a strong recommendation for vaccination is justified. However, I am not sure serological results are really enough justification for the recommendation? Of course in an ideal world they should all be vaccinated, but knowing the practical and financial challenges that this might bring, with potential trade offs in other areas of care, I think this needs careful thought.
Finally, the authors may already be doing this, but given that wild horses live in large groups, it would be really interesting to do the same study in wild horses - not only would this be interesting and relevant in regard of the horses, but it may also support or refute your hypothesis that the reason for your results with donkeys is related to feral donkeys residing in small groups.
I hope the comments are helpful; they are minor points. Otherwise well done in an interesting field of work
Author Response
Thank you very much for your comments. I have included another reference (7) and cited it in lines 226-230 and also lines 43-45. This study looked at PCR results for some of the same pathogens in some of the same captured donkeys after they were co-mingled in holding facilities. In addition we have collected more historic data from the AHDC, including data from domestic donkeys and horses, for a better comparison of serology results. Please also see the revisions made to the conclusion to support further research in the area in order to make any vaccination recommendations.
I would be very interested to do the same study with wild horses--it would compliment this work very nicely.
